# Metabolomic and Proteomic Analyses to Reveal the Role of Plant-Derived Smoke Solution on Wheat under Salt Stress

**DOI:** 10.3390/ijms25158216

**Published:** 2024-07-27

**Authors:** Setsuko Komatsu, Azzahrah Diniyah, Wei Zhu, Masataka Nakano, Shafiq Ur Rehman, Hisateru Yamaguchi, Keisuke Hitachi, Kunihiro Tsuchida

**Affiliations:** 1Faculty of Environment and Information Sciences, Fukui University of Technology, Fukui 910-8505, Japan; 2Hangzhou Institute of Medicine, Chinese Academy of Sciences, Hangzhou 310018, China; zhuwei@him.cas.cn; 3Research Center for Experimental Modeling of Human Disease, Kanazawa University, Kanazawa 920-8640, Japan; masa_nak@staff.kanazawa-u.ac.jp; 4Department of Biology, University of Haripur, Haripur 22620, Pakistan; 5Department of Medical Technology, Yokkaichi Nursing and Medical Care University, Yokkaichi 512-8045, Japan; 6Center for Medical Science, Fujita Health University, Toyoake 470-1192, Japan

**Keywords:** metabolomics, proteomics, salt stress, wheat, plant-derived smoke solution

## Abstract

Salt stress is a serious problem, because it reduces the plant growth and seed yield of wheat. To investigate the salt-tolerant mechanism of wheat caused by plant-derived smoke (PDS) solution, metabolomic and proteomic techniques were used. PDS solution, which repairs the growth inhibition of wheat under salt stress, contains metabolites related to flavonoid biosynthesis. Wheat was treated with PDS solution under salt stress and proteins were analyzed using a gel-free/label-free proteomic technique. Oppositely changed proteins were associated with protein metabolism and signal transduction in biological processes, as well as mitochondrion, endoplasmic reticulum/Golgi, and plasma membrane in cellular components with PDS solution under salt stress compared to control. Using immuno-blot analysis, proteomic results confirmed that ascorbate peroxidase increased with salt stress and decreased with additional PDS solution; however, H^+^-ATPase displayed opposite effects. Ubiquitin increased with salt stress and decreased with additional PDS solution; nevertheless, genomic DNA did not change. As part of mitochondrion-related events, the contents of ATP increased with salt stress and recovered with additional PDS solution. These results suggest that PDS solution enhances wheat growth suppressed by salt stress through the regulation of energy metabolism and the ubiquitin-proteasome system related to flavonoid metabolism.

## 1. Introduction

Plant-derived smoke (PDS) solution is a material for promoting seed germination and plant growth, which affects plant species from various habitats [1]. Butanolides, including karrikins and cyanohydrin, are the main active compounds in PDS solution [2]. Karrikins are the products of smoke released from the heating or combustion of plant material, after which they can stimulate the germination of dormant seeds [3]. Additionally, karrikins have potential functions in mediating abiotic stress tolerance in plants [4,5], and have a similar function to strigolactones in plant adaptation to abiotic stress [6]. PDS solution positively affected the post-germination growth of soybean [7,8], and enhanced soybean growth during and after flooding [9,10]. Furthermore, it enhanced the soybean growth by alleviating salt stress [11]. In the case of wheat, it improved the recovery of plant growth under flooding conditions [12]. Due to PDS solution having numerous components, the positive effects against plant-growth inhibition under abiotic stress are not completely clarified.

Wheat, which supplies approximately 20% of energy and protein requirements, is the third most cultivated cereal crop globally [13]. The increasing frequency and intensity of extreme weather events due to global warming is becoming a problem by reducing wheat yields [14]. Wheat quality and yield are often affected by abiotic stresses including drought and extreme temperatures as well as saline during plant growth and development [15]. As a consequence, it is of great importance for wheat genetic improvements to explore stress tolerance-related genes and screen wheat-germplasm resources with high tolerance [16]. The wheat is sensitive to salt stress, especially high saline soils, which retard its growth as well as delay development, thus causing quality and yield loss [17]. These reports indicated that improving the salt tolerance of wheat is gradually becoming an agricultural research hotspot.

Under salt stress, plants reduce the toxic effects of salt through numerous mechanisms such as antioxidant-enzymes activation, antioxidant-compounds synthesis, ion homeostasis, osmoprotectant biosynthesis, and hormonal regulation [18]. Salt-mediated osmotic stress reduces the ability of plants to draw nutrients and water from the soil, leading to growth stagnation [19]. Ionic toxicity caused by the high concentration of sodium/chloride ions disrupts cellular homeostasis, increases membrane peroxidation, and inhibits photosynthesis in plants [20]. In addition, salt stress induces the accumulation of reactive-oxygen species, such as hydrogen peroxide and superoxide anion, resulting in oxidative damage and/or inhibition of plant growth/development [21]. With regards to these findings, alleviating the pressure of soil salinity, improving plant tolerance to salt stress, and eventually increasing crop yields need to be addressed. 

In this study, a PDS solution is used to characterize salt-tolerant mechanisms in wheat. Because the composition of PDS solution is still not completely understood, its composition was analyzed using liquid chromatography (LC)–mass spectrometry (MS)/MS-based metabolomic analysis. After that, morphological parameters were measured for wheat treated with PDS solution under salt stress. Based on the morphological results, gel-free/label-free proteomics were conducted to explore the tolerant mechanism for the positive effects on the growth of wheat treated with PDS solution under salt stress. Proteomic results were subsequently confirmed by immuno-blot and changes in mRNA were also analyzed by quantitative real time polymerase chain reaction (qRT-PCR).

## 2. Results

### 2.1. Metabolomic Analysis of PDS Solution

Because the composition of PDS solution is still not completely understood, its composition was analyzed using metabolomic technique (Figure 1). After mixing, PDS solution was stored in a refrigerator for one week and upper and lower layers were collected. The upper layer was light brown and the lower layer was dark brown. Plant growth was better with the application of the lower layer of PDS solution than that of the upper layer, even if it was under salt stress (Appendix A). LC–MS/MS-based untargeted metabolomic analysis was separately performed for the upper and lower layers as samples (Appendix A). The metabolomic results of all six samples from the two groups were compared by principal component analysis, which displayed varying accumulation patterns of metabolites between the two different groups (Figure 2). This result indicated that the lower layer of PDS solution largely differs compared with the upper layer (Figure 2). A total of 661 metabolites were kept, with 476 metabolites identified as differentially abundant metabolites with the *p*-value < 0.05 and fold change >1.5 and/or <2/3 (Appendix A). Among them, 32 metabolites increased, while 444 metabolites decreased in the upper layer compared with the lower layer. Among 476 identified metabolites, 340 metabolites were annotated with the Kyoto Encyclopedia of Genes and Genomes (KEGG) IDs and mapped in the KEGG database (Figure 3 and Appendix A). PDS solution, which improves the growth inhibition of wheat under salt stress, contains metabolites associated with flavonoid biosynthesis (Figure 3).

### 2.2. Morphological Analysis of Wheat Treated with PDS Solution under Salt Stress

To investigate the effect of PDS solution on wheat under salt stress, morphological analysis was performed. Three-day-old wheat was treated with or without 2000 ppm PDS solution with 100 mM NaCl for 2 days (Figure 1). Morphological parameters, such as leaf length, fresh leaf weight, main root length, and total fresh root weight were measured (Figure 4). All parameters decreased under salt stress; however, they increased with the application of PDS solution, even if the wheat was under stress (Figure 4). Based on these morphological results, wheat root was used for proteomic analysis.

### 2.3. Identification and Functional Investigation of Proteins in Wheat Treated with PDS Solution under Salt Stress

To investigate the cellular mechanism in wheat growth by the application of PDS solution under salt stress, gel-free/label-free proteomics were conducted. Four kinds of treatments, which were control, salt, PDS, and salt + PDS, were performed. Three-day-old wheat was treated with or without 2000 ppm PDS solution with 100 mM NaCl for 2 days (Figure 1). Proteins extracted from wheat root after treatment were enriched, reduced, alkylated, and digested. After analysis by LC combined with MS/MS, the relative abundance of proteins was compared to each other. In total, 6822 proteins were identified by LC–MS/MS analysis. The proteomic results of all 12 samples from the four groups were compared by principal component analysis, which displayed the varied accumulation patterns of proteins from four different kinds of treatments (Figure 5). This result indicated that salt stress largely affected the wheat proteins; however, this effect was recovered at the protein level by the application of PDS solution, even if it was under stress (Figure 5). 

The abundance of 332 proteins differentially changed with the *p*-value < 0.05 and fold change > 1.5 and/or <2/3 in wheat roots under salt stress compared to the control condition (Appendix A). Among the 332 proteins, 215 and 117 proteins increased and decreased, respectively, under salt stress compared to the control condition (Appendix A and Figure 6 left). On the other hand, the abundance of another 1240 proteins differentially changed with the *p*-value < 0.05 and fold change > 1.5 and/or <2/3 in wheat roots with the PDS solution applied under salt stress compared to the salt stress alone (Appendix A). Among the 1240 proteins, 169 and 1071 proteins increased and decreased, respectively, with the application of the PDS solution under salt stress compared to the salt stress alone (Appendix A and Figure 6 right). The functional category of identified proteins was obtained using gene-ontology analysis (Figure 6). Oppositely changed proteins were associated with protein metabolism and signal transduction in biological processes; mitochondrion, endoplasmic reticulum/Golgi, and plasma membrane in cellular components; and nucleic acid binding activity in molecular mechanisms within the two groups, which are salt/control and salt + PDS/salt (Figure 6). As a result of the functional classification of the proteomic analysis, we focused on categories, which showed opposite trends in increase and decrease. To confirm the results obtained from the proteomic analysis, inversely changed functional categories were further analyzed using immuno-blot and also by qRT-PCR.

### 2.4. Immuno-Blot Analysis of Ascorbate Peroxidase and H^+^-ATPase in Wheat with Application of PDS Solution under Salt Stress

To better uncover the change in proteins from different treatments, immuno-blot analysis of ascorbate peroxidase and H^+^-ATPase in wheat was performed (Figure 7). Because plants have integrated reactive oxygen species detoxification machinery, which includes antioxidant enzymes under salt stress, the behavior of ascorbate peroxidase was confirmed as the positive control. Additionally, H^+^-ATPase was also confirmed because it was significantly changed as a plasma membrane-related protein in proteomic results (Appendix A). Proteins extracted from the root and leaf of wheat were separated on the SDS-polyacrylamide gel by electrophoresis and transferred onto membranes. The membranes were cross-reacted with anti-ascorbate peroxidase and H^+^-ATPase antibodies. A staining pattern with Coomassie brilliant blue was used as a loading control (Appendix A). The integrated densities of bands were calculated using ImageJ software with triplicated immuno-blot results (Appendix A). This analysis confirmed that ascorbate peroxidase increased with salt stress and decreased with additional PDS solution; however, H^+^-ATPase proved opposite effects (Figure 7).

### 2.5. qRT-PCR Analysis of the Gene Encoding Bet v1 in Wheat with Application of PDS Solution under Salt Stress

Bet v1, which is a protein related to signal transduction (Figure 6), significantly increased with salt treatment (Appendix A) and decreased with the application of PDS solution under salt stress (Appendix A). To better uncover the change in proteins from different treatments, qRT-PCR analysis of the gene encoding bet v1 protein in wheat was performed (Figure 8). Based on the proteomic results, the change in the expression level of the gene, which encodes proteins in the response against salt stress, with PDS solution was confirmed with a qRT-PCR analysis. *Bet v1*-specific oligonucleotide was used to amplify transcripts of the total RNA isolated from the wheat root and leaf (Figure 8). The expression level of *18S rRNA* was used as the internal control. The expression of *bet v1* was significantly upregulated by salt stress. Furthermore, following upregulation, the expression recovered to control level with PDS solution, even with stress (Figure 8).

### 2.6. Analyses of Ubiquitin Accumulation and DNA Degradation in Wheat Treated with PDS Solution under Salt Stress

Ubiquitin-related protein, which was categorized as protein metabolism (Figure 6), such as ubiquitin receptor RAD23, significantly increased with salt treatment (Appendix A) and decreased with the application of PDS solution under salt stress (Appendix A). To better understand the change in proteins related to protein degradation from different treatments, the analyses of ubiquitin accumulation and genomic DNA degradation were carried out. Proteins transferred onto membranes were cross-reacted with anti-ubiquitin antibody (Figure 9A and Appendix A). The abundance of ubiquitin in wheat roots increased with salt stress compared to the non-stress condition; and its accumulation was recovered to the level of the non-stressed condition by the treatment of PDS solution even if it was under stress (Figure 9A). The genomic DNA quality in root and leaf was analyzed to verify the occurrence of cell death. Genomic DNA was extracted from root and leaf after 2 days of salt treatment and visualized by agarose-gel electrophoresis (Figure 9B). It did not degrade in wheat in any treatment condition (Figure 9B).

### 2.7. ATP Contents in Wheat with Application of PDS Solution under Salt Stress

Because proteins involved in mitochondrion were oppositely changed in cellular components with or without PDS solution under salt stress (Figure 6), the contents of ATP were analyzed (Figure 10). The contents of ATP increased with salt and recovered with additional PDS solution (Figure 10).

## 3. Discussion

### 3.1. Fraction Containing Metabolites Involved in Flavonoid Biosynthesis Promotes Wheat Growth under Salt Stress

PDS is known to play an important role in the distribution and growth of vegetation [1,22]. A total of 71 compounds were identified in the active fraction of PDS solution by atomic absorption spectrometry and gas chromatography–MS [23]. The isolation of active compounds from PDS solution is difficult due to the large number of compounds present, potentially numbering in the several thousand [23], and due to the low concentration of active compounds compared to the other components [24]. In this study, by fractionating into two fractions, 661 metabolites including new components could be identified (Appendix A). In 2004, 3-methyl-2H-furo [2,3-c]pyran-2-one or karrikinolide was identified as the active compound, which enhanced the germination of many species [25]. Furthermore, PDS containing cyanohydrin glyceronitrile and karrikinolide affected seed germination and plant growth in crops such as maize [26], onion [27], tomato, okura, and beans [28]. However, in this study, karrikin had no effect on wheat growth (Appendix A). This result suggests that other metabolites besides karrikin in PDS solution must be involved in the stress tolerance ability of wheat under salt stress.

Because wheat growth was better with the application of the lower layer in the PDS solution than that of the upper layer, even under salt stress (Appendix A), metabolomic analysis was performed (Figure 2 and Figure 3). Abiotic stress disrupted the pathways of flavonoid biosynthesis in plants, resulting in the interruption of the synthesis of various flavonoids, such as flavonoid, isoflavonoids, and anthocyanins, leading to dysfunction/the loss of function and ultimately affecting the ability of plants to tolerate stress [29,30,31,32]. Flavonoids are secondary metabolites, which play important roles in the tolerance of plants against abiotic stress [33]. Plant hormones have a prominent function in the modulation of the growth, development, reproduction, and secondary metabolism of plants, such as abscisic acid, gibberellic acid, salicylic acid, and jasmonic acid, which are shown to be involved in the regulation of flavonoid biosynthesis [34]. Flavonoid compounds involved in the tolerance of plants against abiotic stresses play key roles in the regulation of various stresses [35,36,37]. In this study, even under salt stress, wheat growth increased with the application of PDS solution, which contains metabolites related to flavonoid biosynthesis (Figure 3 and Figure 4). This result with previous findings suggests that the flavonoid biosynthetic pathway is involved in the PDS solution-induced recovery of growth inhibition in salt-stressed wheat. 

### 3.2. Reactive Oxygen Species Scavenging System Is Related to Salt-Tolerant Mechanism in Wheat Treated with PDS Solution

Plants have integrated reactive oxygen species detoxification machinery, which includes antioxidant enzymes such as ascorbate peroxidase, glutathione reductase, superoxide dismutase, catalase, monodehydroascorbate reductase, and dehydroascorbate, as well as non-enzymatic antioxidants such as ascorbic acid, reduced glutathione, α-tocopherol, carotenoids, phenols, and flavonoids [38]. The application of plant essential oils suggested a possible alternative strategy to increase salt tolerance in durum wheat seedlings towards better growth quality, therefore increasing reactive oxygen species scavenging and the activation of antioxidant defense [39]. The priming imprints of chitosan nanoparticles mitigated the ionic toxicity by upregulating the machinery of antioxidants such as ascorbate peroxidase, glutathione reductase, superoxide dismutase, catalase, flavonoids, and protein contents in wheat seedlings under salt stress [40]. In this study, among proteins related to the scavenging of reactive oxygen species, ascorbate peroxidase increased under salt stress and recovered to control level in wheat treated with PDS solution, even under stress (Figure 4). PDS solution might reduce the accumulation of ascorbate peroxidase by salt stress and improve salt tolerance in wheat.

### 3.3. Energy Metabolism Is Related to Salt-Tolerant Mechanism in Wheat Treated with PDS Solution

Energy metabolism, anthocyanins, photosynthesis, and plant hormones were closely related to the drought resistance of millet and adapt to adversity by precisely regulating the levels of various molecular pathways [41]. When water becomes the limiting factor affecting plant growth, plants sacrifice their growth rate and produce many secondary metabolites such as anthocyanins to protect their growth under adverse conditions [42]. NaCl irrigation increased mineral imbalance, resulting in decreased plant growth, and the levels of most metabolites involved in energy production, sensory quality, and health benefits decreased [43]. The regulation of energy metabolism, protein glycosylation, and cell wall construction was an important factor for the acquisition of salt [11] and flooding tolerance [8] in soybean. In this study, the contents of ATP in wheat increased with salt and recovered to control level with additional PDS solution (Figure 10), which is similar to the previous result [11]. The present result with previous findings suggests that PDS solution may contribute to energy production for the growth of various plants under salt stress.

### 3.4. PDS Solution Relieves Salt Secretion in Wheat under Salt Stress

There are three hypotheses of salt secretion in plants under salt stress, which include (i) the role of the osmotic potential in salt secretion; (ii) a transfer system similar to liquid flow in animals; and (iii) salt solution excretion by vesicles in the plasma membrane [44]. Hydrogen peroxide can be an important signaling molecule in regulating the Na^+^/K^+^ balance. Under salt stress, proteins related to flavonoid biosynthesis, seed storage, and carbohydrate metabolism were identified in the root of *Pongamia pinnata*; indicating that these proteins were most likely recruited from secondary and anaerobic metabolism, which provided defense for roots against Na^+^ toxicity under salt stress [45]. Under salt stress, hydrogen peroxide promotes Na^+^ efflux by stabilizing *SOS1* mRNA, and inhibiting hydrogen peroxide production increases K^+^ efflux [46]. Inhibiting hydrogen peroxide generation and H^+^-ATPase activity altered Na^+^ and K^+^ secretion rates in diploids and tetraploids under salt stress, indicating involvement in regulating Na^+^ and K^+^ secretion [47]. In this study, ascorbate peroxidase increased with salt stress and decreased with additional PDS solution; however, H^+^-ATPase proved opposite effects (Figure 7). These results in addition to previous findings suggest that increased Na^+^ secretion and decreased Na^+^ accumulation in plant tissues mitigate the damage caused by salt stress.

### 3.5. PDS Solution Alleviates Promotion of Cell Death by Salt Stress

Salt stress hinders plant growth and development by impairing vital biological processes such as photosynthesis, energy, and water/nutrient acquisition, which ultimately culminate in cell death when stress intensity becomes uncured [48,49]. Certainly, salt-challenged plants accumulate high inorganic ions, especially Na^+^ and Cl^−^, resulting in cellular toxicity, nutritional/energetic imbalances, and lipid peroxidation. The stress ultimately leads to the production of reactive oxygen species and metabolic dysfunction, which impair photosynthesis and nutrient acquisition, causing cell death based on stress severity [50]. Salinity-induced reactive-oxygen species frequently attack macromolecules in plant cells, resulting in oxidative stress and cell death [51]. Salt stress often induces reactive oxygen species accumulation, leading to membrane lipid peroxidation and cell death [21]. Although ubiquitin increased, DNA degradation did not occur in this study (Figure 9). This result, along with previous findings, suggests that apoptosis is not involved because salt stress promotes protein degradation and PDS solution alleviates it.

## 4. Materials and Methods

### 4.1. Preparation of PDS Solution

PDS solution was prepared from *Cymbopogon jwarncusa* (Kohat University of Science and Technology, Kohat, Pakistan) [9], which was modified from previous methods [52]. Samples were washed with distilled water in order to remove the dust particles and were shade dried. A portion (333 g) of semi-dried plant was smoldered in a furnace which was airtight. Smoke was bubbled through 1 L of distilled water in a beaker to gain concentrated smoke solution which was filtered through sterilized filter paper. Three independent experiments were performed as biological replicates. As independent biological replicates, semi-dried plant was smoldered for different times. After mixing, PDS solution was stored in the refrigerator for one week. Metabolomic analysis was performed separately for the upper and lower layers as samples. 

### 4.2. Metabolomic Analysis 

#### 4.2.1. Metabolites Extraction

Samples (1 mL) were centrifuged at 15,000× *g* for 10 min at 4 °C. The supernatants were transferred to new tubes and dried by lyophilization. They were resuspended in 100 µL of methanol containing 0.1% formic acid, cooled on ice for 5 min, and centrifuged at 15,000× *g* for 10 min at 4 °C. Supernatants (53 µL) were transferred to new tubes and diluted with water containing 53% methanol. They were additionally diluted at 1:100 with 53% methanol containing 0.1% formic acid and transferred to vials for ultra-high performance (UHP)LC–MS analysis [53].

#### 4.2.2. UHPLC–MS Analysis

An Ultimate 3000 UHPLC system (Thermo Fisher Scientific, Waltham, MC, USA) coupled to an Orbitrap Q Exactive Plus (Thermo Fisher Scientific) was used for metabolite analysis. The separation of metabolites using UHPLC conditions was as follows: mobile phase A contained 0.1% formic acid, while mobile phase B contained 99.7% methanol. The extraction solutions were injected into the reverse phase column, Hypersil Gold Column (100 × 2.1 mm with 1.9 µm particle size; Thermo Fisher Scientific). The injection volume was 10 µL, the flow rate was 0.2 mL/min, and the column temperature was 20 °C. Gradient conditions were as follows: 0 min, 98% A and 2% B; 1.5 min, 98% A and 2% B; 12 min, 0% A and 100% B; 14 min, 0% A and 100% B; 14.1 min, 98% A and 2% B; and 17 min, 98% A and 2% B. The Orbitrap Q Exactive Plus MS was operated in positive/negative switching mode with a spray voltage of 3.2 kV. The capillary temperature was 320 °C; the autosampler temperature was 10 °C; and 35 arb and 10 arb were the sheath and auxiliary gas flow rates, respectively [53].

#### 4.2.3. Analysis of Metabolomic Data

Raw data files were converted to analysis base file (ABF) format using Analysis Base File Converter (http://www.reifycs.com/AbfConverter/index.html; 20 January 2021). MS-DIAL5.1 was used for MS data acquisition (https://systemsomicslab.github.io/compms/msdial/main.html; 20 January 2021). A minimum peak height of 1000 and a minimum peak width of 5 were implemented for peak detection, with a sigma window value of 0.5. For metabolite annotation, the MS-DIAL metabolomics MSP spectral kit was integrated and used for MS1 and MS2 annotation. For peak alignment, the RT tolerance was 0.1 min and the retention-time factor was 0.5.

#### 4.2.4. Differential Analysis of Metabolites Using Metabolomic Data

The raw excel data from positive mode and negative modes were annotated with KEGG IDs using Metaboanalyst 6.0 [54] and KEGG database (https://www.genome.jp/kegg/; 1 June 2024). The duplicate entries were removed after combing the negative and positive data. Moreover, metabolites not derived from plants, such as those originating from microbiome, human, drug, and synthetic chemicals based on PubChem, were excluded. The difference between the upper and lower layers was analyzed using Student’s *t*-test in Excel. Furthermore, principal component analysis was performed using Metaboanalyst 6.0. Finally, the results were visualized on a KEGG map.

### 4.3. Plant Material and Treatment

Seeds of wheat (*Triticum aestivum* L. cultivar Nourin 61) were sown in silica sand, which was placed in a seedling case. Three days later, wheat seedlings were treated with or without 2000 ppm PDS solution, as well as with or without 100 mM NaCl. Seedlings were maintained at 25 °C in a growth chamber illuminated with white-fluorescent light (200 µmol m^−2^ s^−1^, 16 h light period/day). For morphological analysis, root and leaf from 5-day-old wheat were used. Regarding the morphological result, root was used for proteomic analysis. Furthermore, root and leaf were used in other biological analyses. More than three independent experiments were performed as biological replicates for all experiments. As independent biological replicates, more than 20 seeds were sown for each treatment for each replicate on different days. 

### 4.4. Protein Extraction

A portion (500 mg) of samples was ground in 500 µL of lysis buffer consisting of 50 mM Tris-HCl (pH 7.6), 100 mM NaCl, 1% Nonidet-P40, 0.1% SDS, and protease inhibitor (Nacalai Tesque, Kyoto, Japan) into a mortar and pestle on ice. The suspension was centrifuged twice at 16,000× *g* for 10 min at 4 °C. Protein concentration was analyzed using XL-Bradford solution (Integrale, Naruto, Japan). After mixing and incubation for 5 min, the absorbance was measured at 595 nm and determined with bovine serum albumin as the standard. 

### 4.5. Proteomic Analysis

#### 4.5.1. Protein Enrichment, Reduction, Alkylation, and Digestion

Quantified proteins (100 µg) were adjusted to a final volume of 100 µL. Proteins were enriched, reduced, alkylated, and digested using previous methods [55]. Briefly, methanol (400 µL) was added to each sample followed by 100 µL of chloroform and 300 µL of water. After centrifugation at 16,000× *g* for 10 min to achieve phase separation, the upper phase was discarded and 300 µL of methanol was added to the lower phase. After centrifugation at 16,000× *g* for 10 min, the pellet was collected as the soluble fraction. Proteins were resuspended in 50 mM ammonium bicarbonate, reduced with 50 mM dithiothreitol for 30 min at 56 °C, and alkylated with 50 mM iodoacetamide for 30 min at 37 °C in the dark. After incubation, proteins were digested with trypsin and lysyl endopeptidase (Wako, Osaka, Japan) at a 1:100 enzyme/protein ratio for 16 h at 37 °C. Peptides were desalted with MonoSpin C18 Column (GL Sciences, Tokyo, Japan) and acidified with 1% trifluoroacetic acid. 

#### 4.5.2. Protein Identification Using Nano LC–MS/MS 

The conditions of LC (EASY-nLC 1000; Thermo Fisher Scientific) and MS (Orbitrap Fusion ETD MS; Thermo Fisher Scientific) were described in the previous study [10]. The peptides were loaded onto the LC system equipped with a trap column (Acclaim PepMap 100 C18 LC column, 3 µm, 75 µm ID × 20 mm; Thermo Fisher Scientific), equilibrated with 0.1% formic acid, and eluted with a linear acetonitrile gradient (0–35%) in 0.1% formic acid at a flow rate of 300 nL min^−1^. The eluted peptides were loaded and separated on the column (EASY-Spray C18 LC column, 3 µm, 75 µm ID × 150 mm; Thermo Fisher Scientific) with a spray voltage of 2 kV (Ion Transfer Tube temperature: 275 °C). The peptide ions were detected using MS in the data-dependent acquisition mode with the installed Xcalibur software (version 4.0; Thermo Fisher Scientific). Full-scan mass spectra were acquired in the MS over 375–1500 *m*/*z* with a resolution of 120,000. The most intense precursor ions were selected for collision-induced fragmentation in the linear ion trap at a normalized collision energy of 35%. Dynamic exclusion was employed within 60 s to prevent the repetitive selection of peptides. 

#### 4.5.3. MS Data Analysis

The MS/MS searches were carried out using MASCOT (version 2.6.2; Matrix Science, London, UK) and SEQUEST HT search algorithms against the UniprotKB *Triticum aestivum* (Wheat4565_SwissProt_TreEMBL_TaxID4546_CanonicalIsoform; version 202102) using Proteome Discoverer 2.4 (version 2.4.1.15; Thermo Fisher Scientific). The workflow was described in the previous study [9]. Both algorithms included spectrum files RC, spectrum selector, MASCOT, SEQUEST HT search nodes, percolator, ptmRS, and minor feature detector nodes. The oxidation of methionine and carbamidomethylation of cysteine were set as a variable modification and a fixed modification, respectively. Mass tolerances in MS and MS/MS were set at 10 ppm and 0.6 Da, respectively. Trypsin was specified as protease and a maximum of 2 missed cleavages was allowed. Target-decoy database searches were used for the calculation of false discovery rate, which was calibrated at 1% for peptide identification.

#### 4.5.4. Differential Analysis of Proteins Using MS Data

Label-free quantification was performed with Proteome Discoverer 2.4 using precursor-ions quantifier nodes. Principal component analysis was performed with Proteome Discoverer 2.4. For the differential analysis of the relative abundance of peptides and proteins between samples, the free software PERSEUS (version 1.6.15.0) [56] was used. The workflow was described in the previous study [9]. The abundances of proteins and peptides were transferred into log2 scale. Three biological replicates of each sample were grouped and a minimum of three valid values were required in one group. The normalization of the abundances was performed to subtract the median of each sample. Missing values were imputed based on a normal distribution (width = 0.3, down-shift = 1.8). Significance was assessed using Student’s *t*-test analysis. The sequences of the differentially accumulated proteins were subjected to a BLAST query against the gene-ontology database (http://www.geneontology.org/; 22 June 2023).

### 4.6. Immuno-Blot Analysis

SDS-sample buffer consisting of 62.5 mM Tris-HCl (pH 6.8), 2% SDS, 5% dithiothreitol, 10% glycerol, and bromophenol blue (Bio-Rad, Hercules, CA, USA) was added to protein samples. Quantified proteins (10 µg) were separated by electrophoresis on a 10% or 12% SDS-polyacrylamide gel and transferred onto a polyvinylidene difluoride membrane using a semidry-transfer blotter. The blotted membrane was blocked for 5 min in Bullet Blocking One regent (Nacalai Tesque). After blocking, the membrane was cross-reacted with the primary antibodies at a 1:1000 dilution for 30 min. As primary antibodies, anti-ascorbate peroxidase [57], H^+^-ATPase (Agrisera, Vannas, Sweden), and ubiquitin (Stressgen Biotechnologies, Sandiego, CA, USA) antibodies were used. Anti-rabbit IgG conjugated with horseradish peroxidase (Bio-Rad) was used as the secondary antibody. After 30 min incubation, signals were detected using TMB solution Western Blotting (Nacalai Tesque). Coomassie brilliant blue staining was used as a loading control. The integrated densities of bands were calculated using Image J software (version 1.53e, Java1.8.0 172, 64 bit; National Institutes of Health, Bethesda, MD, USA). 

### 4.7. RNA Extraction and qRT-PCR Analysis

The samples (500 mg) were snap-frozen in liquid nitrogen and ground into a powder using a mortar and pestle. Total RNA was isolated with RNeasy Plant Mini Kit (Qiagen, Venlo, The Netherlands) according to the protocol from the manufacturer [58]. First-strand cDNA was synthesized from 1 μg of total RNA using the iSuperscript Reverse Transcription Supermix (Bio-Rad). Gene-specific primers were constructed with Primer3Plus software (https://www.bioinformatics.nl/cgi-bin/primer3plus/primer3plus.cgi/; 1 October 2023) [59] and used to amplify the 200–300 bp regions. Gene-specific primers for *18S rRNA* (X02623) (F 5′-TGATTAACAGGGACAGTCGG-3′; R 5′-ACGGTATCTGATCGTCTTCG-3′) and *Bet v1* (A0A3B6TLM4) (F 5′-ACAGCTGGACCCACGAGATC-3′; R 5′-CAGAATCCTTGGCCTTGGTA-3′) were synthesized. Reaction mixture (20 μL) was used to perform qRT-PCR using SYBR Green Supermix (Invitrogen, Waltham, MA, USA). The PCR conditions included an initial step at 95 °C for 30 s, followed by 40 cycles of 95 °C for 10 s, and 60 °C for 30 s. Gene expression was normalized using *18S rRNA* as an internal control. Relative expression was calculated according to the 2^−∆∆ct^ method.

### 4.8. Genomic DNA Extraction and Electrophoresis

A portion (500 mg) of the samples was quickly frozen in liquid nitrogen and ground into a powder with a mortar and pestle. Genomic DNA was extracted using a Genomic DNA Extraction Kit (NucleoSpin Plant II: Macgrey-Nagel, Duren, Germany) following the protocol from the manufacturer [60]. The absorbance of DNA extraction was measured at 260 nm. The genomic DNA was separated by 2% agarose gel and stained with the Atlas ClearSight Gold DNA stain (BioAtlas, Tartu, Estonia). The integrated densities of the bands were calculated using Image J software.

### 4.9. Measurement of ATP Contents

A portion (125 mg) of samples was homogenized in 250 µL of the ATP assay buffer and centrifuged at 16,000× *g* for 10 min at 4 °C. For sample deproteinization and neutralization, the supernatant was treated with a Deproteinizing Sample Preparation Kit (Biovision, Milpitas, CA, USA). Extracts (50 µL) were added to 50 µL of a reaction mixture consisting of ATP converter, ATP probe, ATP developer, and ATP assay buffer in ATP Colorimetric/Fluorometric Assay Kit (Biovision). After mixing and incubation for 30 min at 25 °C in the dark, the absorbance was measured at 570 nm and determined with ATP as the standard.

### 4.10. Statistical Analysis

The statistical significance of data between the 2 groups was analyzed using a Student’s *t*-test. A *p*-value of less than 0.05 was considered statistically significant.

## 5. Conclusions

PDS solution improved the soybean growth even under salt stress and this enhancement was regulated by energy metabolism, protein glycosylation, and cell wall construction [11]. In this study, to investigate the salt-tolerant mechanism of wheat by PDS solution, metabolomic and proteomic techniques were used. The main findings of this research are as follows: (i) a PDS solution with the activity of wheat growth under salt stress contains metabolites related with flavonoid biosynthesis; (ii) oppositely changed proteins were associated with protein metabolism and signal transduction in biological processes as well as with mitochondrion, endoplasmic reticulum/Golgi, and plasma membrane in cellular components with or without PDS solution under salt stress; (iii) immuno-blot analysis confirmed that ascorbate peroxidase increased with salt stress and decreased with additional PDS solution but H^+^-ATPase demonstrated opposite effects; (iv) ubiquitin increased with salt stress and decreased with additional PDS solution, but genomic DNA did not change; and (v)ATP contents increased with salt stress and recovered with additional PDS solution. These results suggest that PDS solution including the components related to flavonoid metabolism improves wheat growth by alleviating salt stress. Additionally, salt stress in wheat might be associated with the regulation of energy metabolism and the ubiquitin-proteasome system.

## Figures and Tables

**Figure 1 ijms-25-08216-f001:**
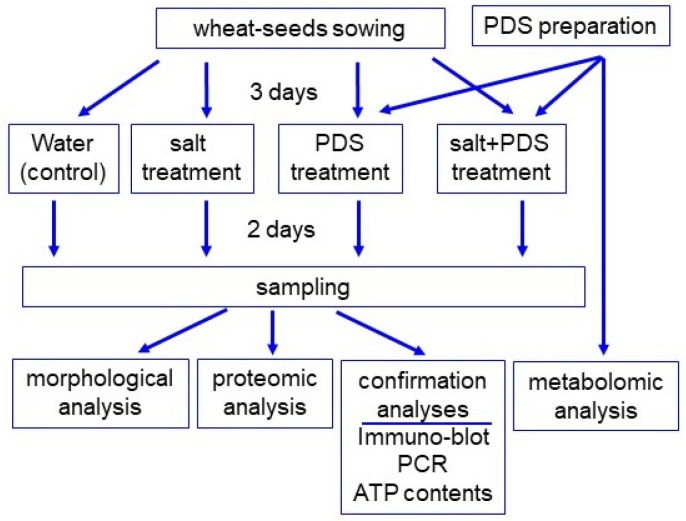
The experimental design for the investigation of the effect of PDS solution on wheat under salt stress. To investigate the potential effects of PDS solution on wheat, its metabolite contents were analyzed using metabolomic methods. After 3 days of sowing, wheat was treated with or without 2000 ppm PDS solution (PDS) with or without 100 mM NaCl (salt) for 2 days. Wheat seedlings were analyzed with morphological and proteomic methods, and confirmation. For confirmation experiments, immuno-blot, PCR, and enzymatic analyses were used. All experiments were performed with three independent biological replicates.

**Figure 2 ijms-25-08216-f002:**
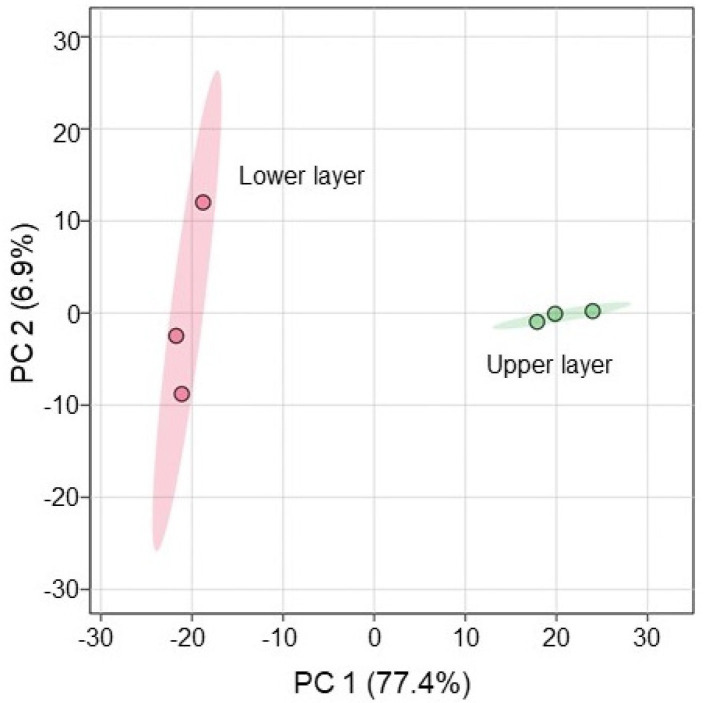
Overview of total metabolomic data from six samples of PDS solution based on principal component analysis. Metabolomic analysis was performed with three independent biological replicates for the lower layer and the upper layer of PDS solution. Principal component analysis was performed with MetaboAnalyst 6.0.

**Figure 3 ijms-25-08216-f003:**
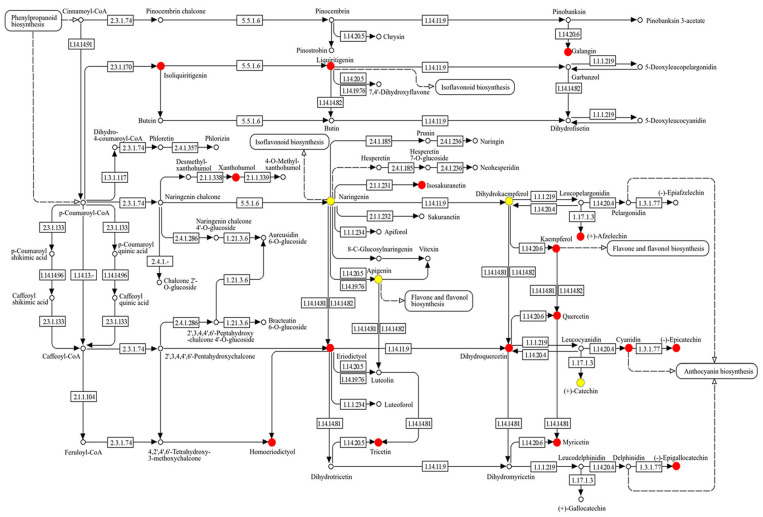
KEGG map of flavonoid biosynthesis-related metabolites changed in the lower layer compared with the upper layer of PDS solution. Metabolites, which were significantly changed between the lower layer and the upper layer of PDS solution were analyzed using a KEGG map (Appendix A). Among them, flavonoid biosynthesis-related metabolites are highlighted. Red color shows significantly increased metabolites and yellow color shows the identified metabolites that were not significantly changed. The numbers in the figure show the EC number.

**Figure 4 ijms-25-08216-f004:**
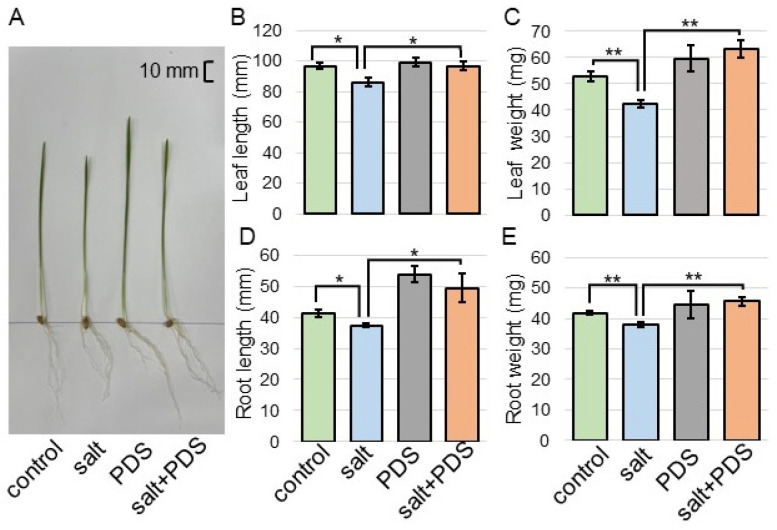
Morphological analysis of wheat treated with PDS solution under salt stress. Three-day-old wheat was treated with or without 2000 ppm PDS solution with or without salt stress for 2 days. The bar in the left panel indicates 10 mm in the picture (**A**). As morphological parameters, leaf length (**B**), fresh leaf weight (**C**), main root length (**D**), and total root fresh weight (**E**) were analyzed at 5 days after sowing. The data are presented as mean ± SD from three independent biological replicates. As independent biological replicates, more than 20 seeds were sown for each treatment for each replicate on different days. Student’s *t*-test was used to compare values between two groups. Asterisks indicate significant changes (*, *p* ≤ 0.05; **, *p* ≤ 0.01).

**Figure 5 ijms-25-08216-f005:**
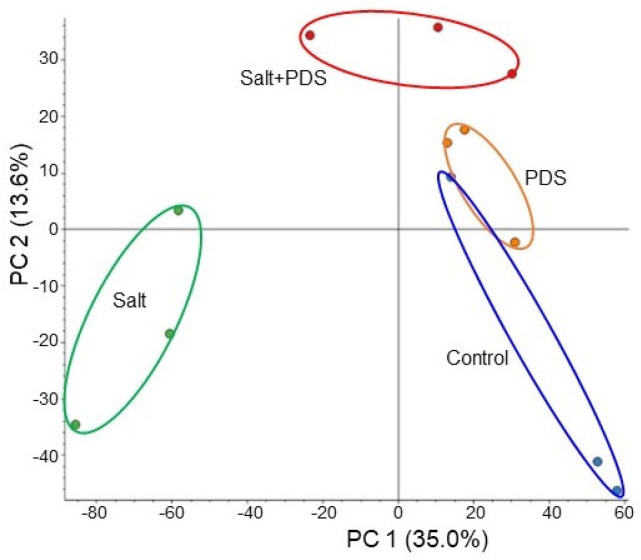
Overview of total proteomic data from 12 samples of wheat based on principal component analysis. Three-day-old wheat was exposed without or with 2000 ppm PDS solution under 100 mM NaCl for 2 days. Proteomic analysis was performed with three independent biological replicates for each treatment. Principal component analysis was performed with Proteome Discoverer 2.4.

**Figure 6 ijms-25-08216-f006:**
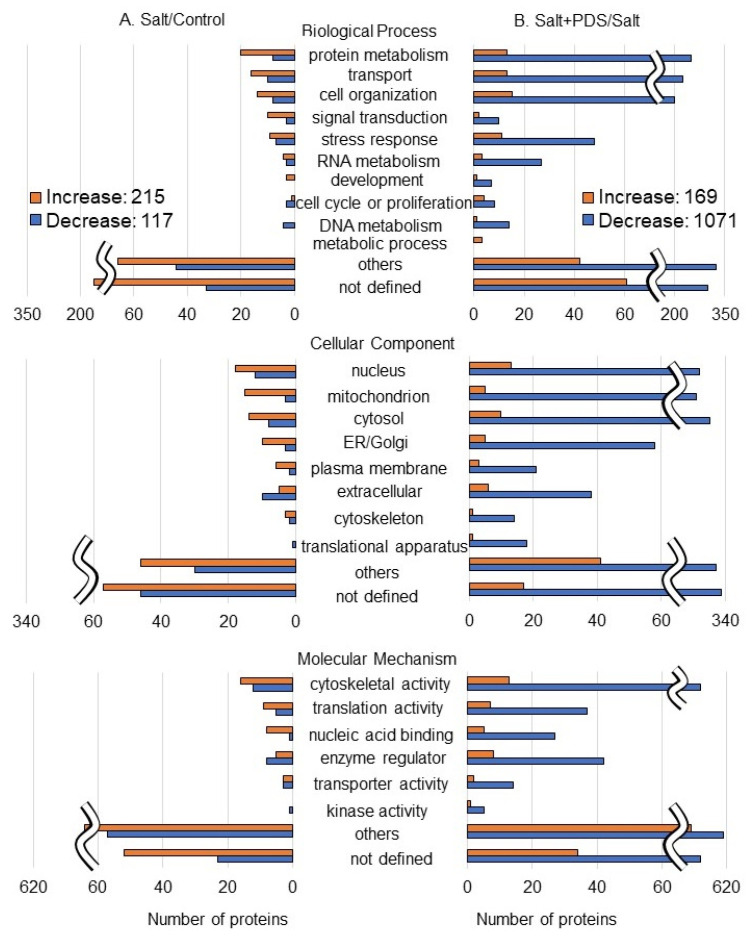
Functional categories of proteins with differential abundance in wheat root with PDS solution under salt stress. Four kinds of treatments, which were control, salt, PDS, and salt + PDS, were performed. Proteins extracted from wheat root after treatment were enriched, reduced, alkylated, and digested. After analysis by LC–MS/MS, the relative abundance of proteins from wheat was compared as follows: salt/control (**A**) (Appendix A) and salt + PDS/salt (**B**) (Appendix A). Functional categories of changed proteins were determined using gene-ontology analysis. Red and blue columns show increased and decreased proteins, respectively.

**Figure 7 ijms-25-08216-f007:**
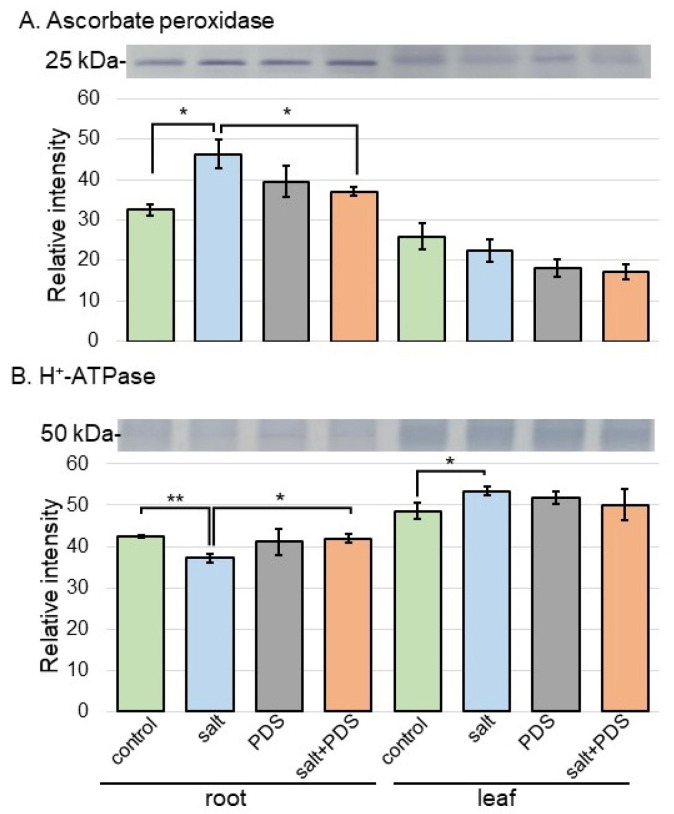
Immuno-blot analysis of the proteins involved in wheat treated with PDS solution under salt stress. Proteins extracted from wheat root and leaf were separated on 10% SDS-polyacrylamide gel by electrophoresis and transferred onto membranes. The membranes were cross-reacted with anti-ascorbate peroxidase (**A**) and H^+^-ATPase (**B**) antibodies. A staining pattern with Coomassie brilliant blue was used as a loading control (Appendix A). The integrated densities of the bands were calculated using ImageJ software. The data are presented as mean ± SD from three independent biological replicates (Appendix A). Student’s *t*-test was used to compare values between two groups. Asterisks indicate significant changes (*, *p* ≤ 0.05; **, *p* ≤ 0.01).

**Figure 8 ijms-25-08216-f008:**
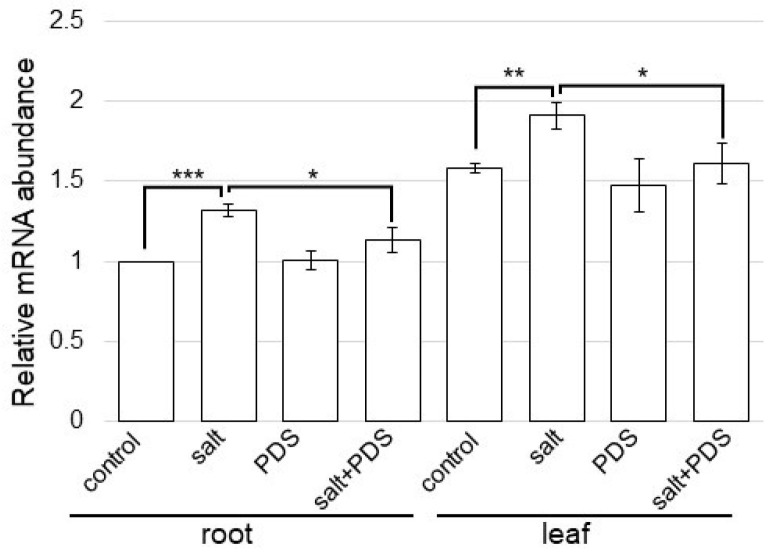
Gene expression of *bet v1* in wheat treated with PDS solution under salt stress. After extraction of total RNA from root and leaf, gene expression analysis of *bet v1* was performed using qRT-PCR. *18S rRNA* was used as an internal control. Data are shown as the means ± SD from three independent biological replicates. Student’s *t*-test was used to compare values between two groups. Asterisks indicate significant changes (*, *p* ≤ 0.05; **, *p* ≤ 0.01; and ***, *p* ≤ 0.001).

**Figure 9 ijms-25-08216-f009:**
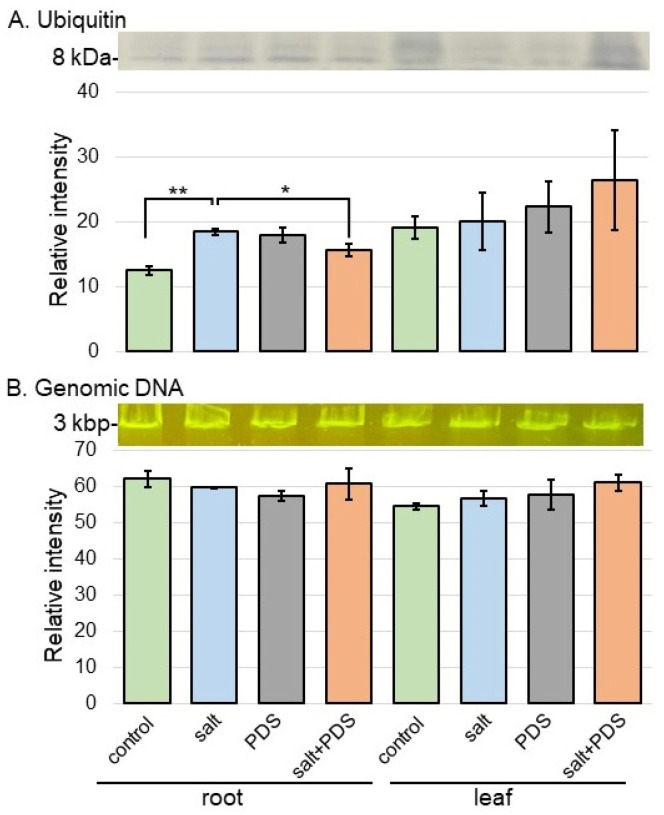
Ubiquitin accumulation and DNA degradation in wheat treated with PDS solution under salt stress. (**A**) Proteins were separated by electrophoresis on a 12% SDS-polyacrylamide gel. Proteins blotted on the membrane were cross-reacted with anti-ubiquitin antibody. A staining pattern with Coomassie brilliant blue was *used* as a loading control (Appendix A). The integrated densities of the bands were calculated using ImageJ software. The data are presented as mean ± SD from three independent biological replicates (Appendix A). (**B**) Genomic DNA degradation in wheat treated with PDS solution under salt stress. After salt treatment, genomic DNA was extracted from root and leaf. Extracted genomic DNA was analyzed by agarose-gel electrophoresis and stained. The integrated densities of bands were calculated using ImageJ software. Data are shown as mean ± SD from three biological replicates. Student’s *t*-test was used to compare values between two groups. Asterisks indicate significant changes (*, *p* ≤ 0.05; **, *p* ≤ 0.01).

**Figure 10 ijms-25-08216-f010:**
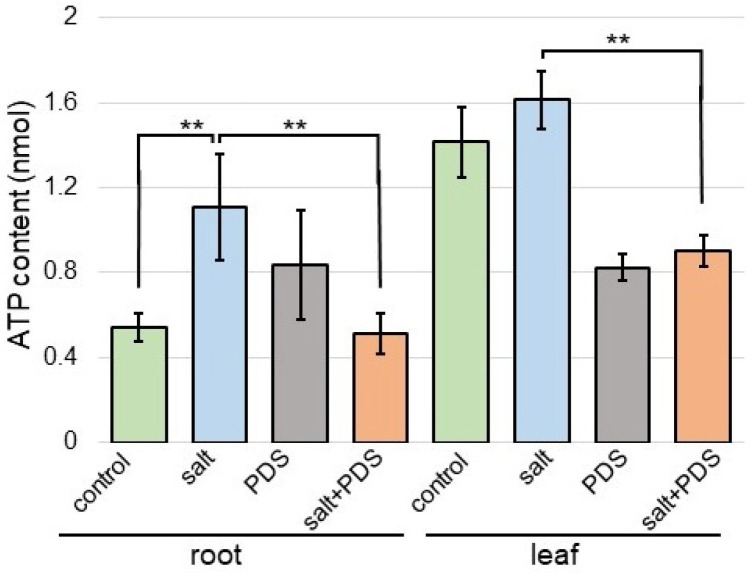
The contents of ATP in wheat treated with the PDS solution under salt stress. Wheat seeds were sown and treated with or without 2000 ppm PDS solution. Three-day-old wheat was treated with or without salt stress for 2 days. Metabolites were extracted from the root and leaf. The ATP contents were measured for each sample. The data are given as the mean ± SD from three independent biological replicates. Student’s *t*-test was used to compare values between two groups. Asterisks indicate significant changes (**, *p* ≤ 0.01).

## Data Availability

MS data, RAW data, peak lists, and result files of proteomic analysis have been deposited in the ProteomeXchange Consortium [61] via the jPOST [62] partner repository under data-set identifiers PXD043228. MS data, RAW data, peak lists, and result files of metabolomic analysis have been deposited in the MetaboBank of DDBJ [63] under accession code MTBKS244.

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
