# Peer review of "Metabolomic and Proteomic Analyses to Reveal the Role of Plant-Derived Smoke Solution on Wheat under Salt Stress"

_ijms, 2024, doi:10.3390/ijms25158216_

Round 1

Reviewer 1 Report

Comments and Suggestions for Authors

The study is original, but the way it has been conducted raise many doubts on the validity and reproducibility of the results. There are several essential questions that need to be adressed before being able to give a decission.

The first thing is the great difference among the upper layer and the lower layer (figure 2). Wouldn't it have been easy to homogenize the PDS (shaking) and use an homogeneous solution in order to grant reproducibility of the results? How easy is to distinguish among both layers? visually? It depends to the distance to the top or to the bottom of the flask? In addition, according to the methods section it looks like the PDS was prepared by the authors and all the tests were conducted with a single batch. How confident are the authors that with a different batch they are going to have similar results.

Figure 4: differences are minimal and the n number is very low (3), but surprisingly authors obtain significant differences (!!). After three days there is still many variability among seedlings, mainly due to minimal differences in the germination timing. Have the auhtors checked whether the effect of PDS is mainteined with older plants and with high number of plants, were the timing differences have been neutralized?

Figure 6: according to this there are about 169 up regulated proteins and more than 1000 down regulated, encompassing many functional categories. So then, why authors decide to follow the study with standards proteins for salt stress and for plant physiology. The main question here is: which is the useful information derived from the proteomic analysis? It is not clear the relation with figure 6 and the subsequent analysis with very obvious proteins? Is there any new information from the proteomics data?

From the technical point of view some experiments are very poor. For instance: 

Figure 8 and supplementary blot figure 5: semiquantitative cDNA PCR is not enough informative. Please perform a realtime quantitative PCR.

Supplementary blot S6 and figure 9. if you want to track ubiquitin, use a gel with higher acrylamide concentration, at that part of the gel, is not resolving small proteins. 

The abstract needs to be rewritten as the use of english is difficult to understand, as the sentence use strange grammar constructions.

With so many weak points paper is not suitable for publication.

Comments on the Quality of English Language

Paper requires a thorough revision

Author Response

Reviewer 1

The study is original, but the way it has been conducted raise many doubts on the validity and reproducibility of the results. There are several essential questions that need to be adressed before being able to give a decission.

Answer: Thank you very much for your suggestion. This article has been carefully revised with additional experiments. Your valuable comments have improved the content of this paper.

The first thing is the great difference among the upper layer and the lower layer (figure 2). Wouldn't it have been easy to homogenize the PDS (shaking) and use an homogeneous solution in order to grant reproducibility of the results? How easy is to distinguish among both layers? visually? It depends to the distance to the top or to the bottom of the flask? In addition, according to the methods section it looks like the PDS was prepared by the authors and all the tests were conducted with a single batch. How confident are the authors that with a different batch they are going to have similar results.

Answer: Thank you very much for your comments and suggestion. Based on the comments and suggestion from reviewers, the section of “Results” has been modified in red. In terms of color, the upper layer was light brown and the lower layer was dark brown. The section “Materials and Methods” has been corrected as follows: “Three independent experiments were performed as biological replicates. As independent biological replicates, semi-dried plant was smoldered in different time. After mixing, PDS solution was stored in the refrigerator for one week. Metabolomic analysis was performed separately for the upper and lower layers as samples.”

Figure 4: differences are minimal and the n number is very low (3), but surprisingly authors obtain significant differences (!!). After three days there is still many variability among seedlings, mainly due to minimal differences in the germination timing. Have the auhtors checked whether the effect of PDS is mainteined with older plants and with high number of plants, were the timing differences have been neutralized?

Answer: We are sorry that these explanations were not clear. For morphological experiments, more than three independent experiments were performed as biological replicates including preliminary experiments. As independent biological replicates, more than 20 seeds were sown for each treatment for each replicate on different days. “4.3. Plant Material and Treatment” has been re-written as follows in red: “More than 3 independent experiments were performed as biological replicates for all experiments. As independent biological replicates, more than 20 seeds were sown for each treatment for each replicate on different days.”

Figure 6: according to this there are about 169 up regulated proteins and more than 1000 down regulated, encompassing many functional categories. So then, why authors decide to follow the study with standards proteins for salt stress and for plant physiology. The main question here is: which is the useful information derived from the proteomic analysis? It is not clear the relation with figure 6 and the subsequent analysis with very obvious proteins? Is there any new information from the proteomics data?

Answer: We are sorry for this problem. As you suggested, the relationship between figure 6 and the subsequent analysis has been described in the Results section in red. As a result of functional classification of the proteomic analysis, we focused to categories, which showed opposite trends in increase and decrease. We then conducted detailed verification experiments on proteins in those categories, which showed significant increases and decreases. A novel result is the detection of proteins involved in flavonoid synthesis and metabolism. These findings have been added in the Discussion section in red.

From the technical point of view some experiments are very poor. For instance: 

Figure 8 and supplementary blot figure 5: semiquantitative cDNA PCR is not enough informative. Please perform a realtime quantitative PCR.

Answer: As suggested, additional experiment with qRT-PCR has been performed. Based on this result, Figure 8, Result section “2.5”, and Materials and Methods “4.7” have been revised in red.

Supplementary blot S6 and figure 9. if you want to track ubiquitin, use a gel with higher acrylamide concentration, at that part of the gel, is not resolving small proteins. 

Answer: As suggested, additional experiment with higher acrylamide concentration has been performed. Based on this result, Figure 9, Figure S6, and Materials and Methods “4.6” have been revised in red.

The abstract needs to be rewritten as the use of english is difficult to understand, as the sentence use strange grammar constructions.

Answer: This manuscript was revised by a native English speaker. However, this manuscript has been reviewed again following comments from reviewer 1. The revisions have been marked in red.

With so many weak points paper is not suitable for publication.

Answer: Thank you very much for your critical comments. Based on the comments, this article has been revised with additional experiments. Your valuable comments have improved the content of this paper.

Reviewer 2 Report

Comments and Suggestions for Authors

The authors identify key metabolites and proteins affected by plant-derived smoke (PDS) under salt stress using metabolomic and proteomic analyses, suggesting that PDS solution improves wheat growth by  regulating energy metabolism and ubiquitin-proteasome system.  The paper presents valuable research on a significant agricultural issue, contributing to the broader understanding of plant stress responses. However, it lacks detailed experimental information and comparative analysis, which could enhance the robustness and applicability of the findings.

(1) The paper lacks detailed information on the experimental setup, such as the concentration and duration of PDS solution application, which are crucial for reproducibility and further research. Can the authors provide more detailed information on the concentration and duration of PDS  solution application in their experiments?

(2) The paper does not provide a thorough comparison with widely-known stress-tolerance mechanisms or treatments,  which would strengthen the argument for the novelty and effectiveness of PDS solution. Could the authors compare their findings with other known stress-tolerance mechanisms or treatments to highlight the unique contributions of PDS solution?

(3)The evaluation of the proposed mechanism is limited to a single species (wheat), and  it would be beneficial to include comparative studies with other plants to generalize the findings. Are there any plans to extend the study to other plant species to validate  the generalizability of the findings?

Author Response

Reviewer 2

The authors identify key metabolites and proteins affected by plant-derived smoke (PDS) under salt stress using metabolomic and proteomic analyses, suggesting that PDS solution improves wheat growth by  regulating energy metabolism and ubiquitin-proteasome system.  The paper presents valuable research on a significant agricultural issue, contributing to the broader understanding of plant stress responses. However, it lacks detailed experimental information and comparative analysis, which could enhance the robustness and applicability of the findings.

Answer: Thank you very much for your suggestion. This article has been carefully revised with additional experiments. Your valuable comments have improved the content of this paper.

(1) The paper lacks detailed information on the experimental setup, such as the concentration and duration of PDS solution application, which are crucial for reproducibility and further research. Can the authors provide more detailed information on the concentration and duration of PDS  solution application in their experiments?

Answer: We are sorry for this problem. The concentration and duration of PDS-solution application has been specified in each part in red. In the materials and methods “4.3 Plant Material and Treatment”, it has been clarified in red as follows: “Seeds of wheat (Triticum aestivum L. cultivar Nourin 61) were sown in silica sand, which was placed in seedling case. Three days later, wheat seedlings were treated with or without 2000 ppm PDS solution, as well as with or without 100 mM NaCl.”

(2) The paper does not provide a thorough comparison with widely-known stress-tolerance mechanisms or treatments,  which would strengthen the argument for the novelty and effectiveness of PDS solution. Could the authors compare their findings with other known stress-tolerance mechanisms or treatments to highlight the unique contributions of PDS solution?

Answer: Thank you very much for your suggestion and we are sorry for this problem. As suggested, because it has generally been shown that the reactive-oxygen species scavenging system is involved under salt stress, it has been discussed. The comparison between this study and previous reports has been added with some information in new paragraph “3.2 Reactive-Oxygen Scavenging System Is Related to Salt-Tolerant Mechanism in Wheat Treated with PDS Solution”.

(3)The evaluation of the proposed mechanism is limited to a single species (wheat), and  it would be beneficial to include comparative studies with other plants to generalize the findings. Are there any plans to extend the study to other plant species to validate  the generalizability of the findings?

Answer: Thank you very much for your valuable comments. We have already analyzed the salt-tolerant mechanism of PDS solution using soybean. Article is “Komatsu, S.; Kimura, T.; Rehman, S.U.; Yamaguchi, H.; Hitachi, K.; Tsuchida, K. Proteomic analysis reveals salt-tolerant mechanism in soybean applied with plant-derived smoke solution. Int. J. Mol. Sci. 2023, 24, 13734.” Based on the suggestion, the comparison between soybean and wheat has been added in “Introduction” and “Discussion” as well as “Conclusion” in red.

Round 2

Reviewer 1 Report

Comments and Suggestions for Authors

Authors have made a great effort in improving the manuscript. Now is more coherent and most of my concerns have been properly adressed. I can recommend publication.